# Secure image communication based on two-layer dynamic feedback encryption and DWT information hiding

**Jinlong Zhang**[1], **Heping Wen**[ORCID][2,3] *

**1** School of Information Technology and Management, Hunan University of Finance and Economics, Changsha, China, **2** Zhongshan Institute, University of Electronic Science and Technology of China, Zhongshan, China, **3** University of Electronic Science and Technology of China, Chengdu, China

* wenheping@uestc.edu.cn

**Data Availability Statement:** All relevant data are within the paper.

**Funding:** The author(s) received no specific funding for this work.

## Abstract

In response to the vulnerability of image encryption techniques to chosen plaintext attacks, this paper proposes a secure image communication scheme based on two-layer dynamic feedback encryption and discrete wavelet transform (DWT) information hiding. The proposed scheme employs a plaintext correlation and intermediate ciphertext feedback mechanism, and combines chaotic systems, bit-level permutation, bilateral diffusion, and dynamic confusion to ensure the security and confidentiality of transmitted images. Firstly, a dynamically chaotic encryption sequence associated with a secure plaintext hash value is generated and utilized for the first round of bit-level permutation, bilateral diffusion, and dynamic confusion, resulting in an intermediate ciphertext image. Similarly, the characteristic values of the intermediate ciphertext image are used to generate dynamically chaotic encryption sequences associated with them. These sequences are then employed for the second round of bit-level permutation, bilateral diffusion, and dynamic confusion to gain the final ciphertext image. The ciphertext image hidden by DWT also provides efficient encryption, higher level of security and robustness to attacks. This technology offers indiscernible secret data insertion, rendering it challenging for assailants to spot or extract concealed information. By combining the proposed dynamic closed-loop feedback secure image encryption scheme based on the 2D-SLMM chaotic system with DWT-based hiding, a comprehensive and robust image encryption approach can be achieved. According to the results of theoretical research and experimental simulation, our encryption scheme has dynamic encryption effect and reliable security performance. The scheme is highly sensitive to key and plaintext, and can effectively resist various common encryption attacks and maintain good robustness. Therefore, our proposed encryption algorithm is an ideal digital image privacy protection technology, which has a wide range of practical application prospects.

**Competing interests:** The authors have declared that no competing interests exist.

## Introduction

In recent years, the advancements in computer communication and network technologies have facilitated the frequent, widespread, and rapid dissemination of various types of data and information through networks [1–3]. As a result, there has been a growing need for a secure transmission environment to accommodate the diverse forms of data exchanged [4–6]. Among these, images stand out as a highly visual and commonly shared form of data that typically contains a significant amount of private and sensitive information [7–9]. Therefore, the application of image encryption techniques has emerged as a valuable approach to safeguarding important data from potential leakage during transmission [10–12]. By employing such techniques, the confidentiality and integrity of transmitted images can be effectively maintained [13–15], ensuring the preservation of privacy and the mitigation of unauthorized access or malicious attacks [16–18]. Numerous encryption techniques have been suggested, such as chaos theory [19–21], frequency domain encryption [22–24], bit-level coding [25–27], DNA coding [28–30], compressed sensing [31–33], thumbnail-preserving encryption [34, 35] and so on [2, 36, 37]. Furthermore, the utilization of chaos in image encryption algorithms has gained immense popularity due to its inherent characteristics of unpredictability, pseudo-randomness, and sensitivity to initial values [38, 39]. These properties make chaos not only highly effective but also exceptionally suitable for ensuring the security and confidentiality of image data. By leveraging the unpredictable and sensitive nature of chaos, image encryption algorithms can provide robust protection against potential attacks or unauthorized access, making them a preferred choice in the field of image security.

Throughout the international status, many researchers have attained a range of significant theoretical [40–42] and practical outcomes [43–45] when employing chaotic system for image encryption. In 2022, Ref. [36] introduces an image encryption scheme that relies on a memristive chaotic system, combining cyclic shift and dynamic DNA-level diffusion. Simulation tests and analyses show that this scheme is highly secure and resistant to various attacks. In 2023, Ref. [46] presented a dynamic RNA-encoded color image encryption scheme that utilizes a chain feedback structure. The color image is encrypted using a chaotic sequence based on plaintext correlation for each color component and the color-coded image is obtained through RNA dynamic encoding and other operations. The results of the experiment demonstrate that the encryption algorithm exhibits outstanding encryption effectiveness and security performance in the face of different attacks [47–49]. In the realm of chaotic image encryption research, the performance of chaotic systems and algorithms significantly influences the security and efficiency of cryptographic systems. It is particularly important and urgent to explore an image encryption algorithm that uses chaotic mapping constructs to resist various illegal attacks.

In this paper, a dynamic closed-loop feedback secure image encryption scheme based on 2D-SLMM chaotic system is suggested. The algorithm uses the chaotic encryption sequence associated with the plaintext image to perform the first round of encryption, and then uses the chaotic sequence associated with the intermediate ciphertext to perform the second round of encryption. A color image encryption scheme based on chaotic system, bit-level permutation, bilateral diffusion and dynamic confusion is proposed. The experimental findings demonstrate that the algorithm exhibits exceptional encryption effectiveness and efficient encryption performance. Furthermore, the suggested image encryption algorithm can effectively withstand various unauthorized attacks.

The main innovations and contributions of this paper can be outlined as follows.

- Image encryption using DWT hiding method offers efficient encryption, a higher level of security, and robustness against attacks. DWT-based hiding allows for effective encryption

by decomposing the image into frequency sub-bands, enabling secret data to be hidden in high-frequency sub-bands without compromising the visual quality in low-frequency sub-bands. This technique provides indistinguishable embedding of secret data, making it challenging for attackers to detect or extract the hidden information. Additionally, DWT-based hiding ensures the integrity of embedded secret data even when the encrypted image is subjected to common attacks like noise addition or compression. Overall, DWT hiding is an effective approach for securely protecting sensitive information in images while preserving their visual quality.

- A majority of the current encryption methodologies rely on pixel-level encryption, which offers inadequate encryption granularity, thereby posing a security risk. However, this secure image encryption algorithm addresses these issues by utilizing bit-level permutation, bilateral diffusion, and dynamic confusion. It combines various encryption technologies, such as chaotic encryption, to enhance the security and robustness of the algorithm. As a result, it effectively improves the security, making it more resistant to attack and ensuring the confidentiality of the transmitted image.

- Numerous existing encryption algorithms feature flawed structures. In the absence of plaintext correlation or ciphertext feedback, they are susceptible to known plaintext attacks or chosen plaintext attacks. This secure image encryption scheme uses a dynamic feedback mechanism to update the encryption key according to the encrypted data. This feedback mechanism enhances the security and resistance of attacks such as chosen plaintext attacks and chosen ciphertext attacks.

- The existing low-dimensional chaotic systems have the risk of being estimated or identified due to their insufficient security. The color image encryption algorithm uses a 2D-SLMM chaotic system combining Logistic map and sine map to generate pseudo-random sequences for encryption. The 2D-SLMM system enhances the nonlinearity and randomness of the encryption process and ensures the confidentiality and integrity of the transmitted image.

The rest of this article is organized as follows. Section 1 briefly introduces the 2D-SLMM chaotic system and the related theory of dynamic image eigenvalue extraction. Section 2 presents the encryption algorithm designed in this paper. In section 3, the experimental results and simulation results are given. The last part is the conclusion of the paper.

## Related theories

### The used chaotic system

Chaotic systems find extensive application in image encryption algorithms due to their notable features, including high sensitivity to initial values, non-divergence, non-convergence, non-periodicity, and the consequent generation of highly random sequences. The chaotic system used in this paper is 2D Sine Logistic Modulation Map, which can be defined by the following Eq (1):

$$\begin{cases} h_{i+1} = a(\sin(\pi w_i) + b)x_i(1 - h_i) \\ w_{i+1} = a(\sin(\pi h_i) + b)w_i(1 - w_i) \end{cases} \tag{1}$$

where $a$ and $b$ are control parameters. When $a \in [0, 1]$ and $b \in [0, 3]$, the system is in a chaotic state. The Logistic map and the Sine map are nonlinear transformations of simple structures, so their orbits are easy to predict. The 2D-SLMM used in this paper is derived from logical mapping and sinusoidal mapping. The integration of sine mapping with the parameter $b$ is

employed to modulate the output of the Logistic mapping, thereby enhancing its nonlinearity and randomness. Subsequently, these outcomes are extended from the one-dimensional domain to the two-dimensional domain to create the 2D-SLMM system.

**NIST test.**  The NIST 800-22 test suite serves as a comprehensive statistical toolbox comprising 16 distinct tests meticulously designed for the assessment of random binary sequences. These sequences can vary in length, making the suite an invaluable tool for evaluating the output of both hardware and software-based cryptographic random or pseudo-random number generators.

In this particular evaluation, all the sequences employed in encryption processes have successfully met the rigorous criteria set by the NIST 800-22 test suite. The meticulous examination of these sequences resulted in a series of partial test results, which are thoughtfully summarized in Table 1. This achievement signifies the robustness and reliability of the cryptographic random or pseudorandom number generators utilized in the encryption procedures.

It is worth noting that such comprehensive testing procedures are critical in the realm of cryptography, where the unpredictability and statistical quality of random sequences are of paramount importance. The successful passage of these tests not only provides confidence in the cryptographic system's security but also assures users that the generated random or pseudorandom numbers adhere to the high standards set by NIST, reinforcing the integrity of the encryption process.

## Discrete wavelet transform

The DWT is a powerful mathematical tool used in signal processing and image compression. It decomposes a signal or an image into different frequency components, allowing for both time and frequency domain analysis. DWT is widely applied in various fields, including image processing, data compression, and denoising.

The wavelet transform operates by iteratively refining the signal across multiple scales, achieving this through a series of scaling and translation operations. This progressive refinement process culminates in a remarkable outcome: a high-frequency time division and a low-frequency frequency division of the signal. This unique characteristic enables the wavelet

**Table 1. NIST-800-22 test results.**

| Statistical Tests | *p*-Values Sequence | Results |
|---|---|---|
| Frequency (Monobit) Test | 0.213309 | successful |
| Block-Frequency Test | 0.534146 | successful |
| Cumulative-Sums Test | 0.122325 | successful |
| Runs Test | 0.534146 | successful |
| Longest-Run Test | 0.534146 | successful |
| Binary Matrix Rank Test | 0.911413 | successful |
| Discrete Fourier Transform Test | 0.350485 | successful |
| Non-Overlapping Templates Test | 0.004301 | successful |
| Overlapping Templates Test | 0.366918 | successful |
| Maurer's Universal Statistical Test | 0.534146 | successful |
| Approximate Entropy Test | 0.051942 | successful |
| Random-Excursions Test ($x = -4$) | 0.022503 | successful |
| Random-Excursions Variant Test ($x = -9$) | 0.022503 | successful |
| Serial Test-1 | 0.637119 | successful |
| Serial Test-2 | 0.040108 | successful |
| Linear-Complexity Test | 0.224821 | successful |

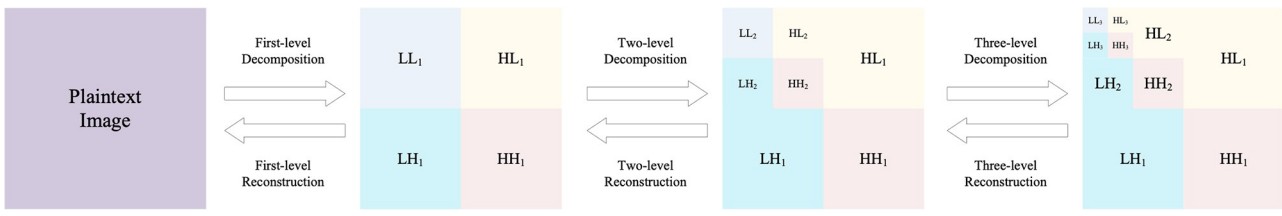

**Fig 1. Block diagram of DWT program.**

transform to automatically adapt to the intricate demands of time-frequency signal analysis. The schematic diagram of the image wavelet decomposition is shown in Fig 1. The DWT can be represented by Eq (2) as:

$$W(e, q) = \sum_{n=0}^{N-1} x(n) \cdot \psi_{e,q}(n) \qquad (2)$$

where $W(e, q)$ represents the transformed coefficient, with $a$ and $b$ denoting the scale and translation parameters, respectively. These parameters are utilized to control the shape and position of the wavelet function. $x(n)$ corresponds to the discrete sample values of the input signal. $\psi_{e,q}(n)$ represents the wavelet function, which is dependent on the scale parameter $e$ and translation parameter $q$.

## The proposed encryption algorithm

Compared with the traditional algorithm, this paper adopts the plaintext correlation mechanism, and uses the MD5 hash function to make the key dynamically update with the encrypted data. Using the key, the sequence is generated by 2D-SLMM chaos. The sequence is used to perform bit-level permutation, bilateral diffusion and dynamic confusion on the image. And innovatively use the ciphertext feedback method to perform two-round encryption, so that the algorithm not only has computational complexity, but also has the ability to resist the chosen plaintext attacks and the chosen ciphertext attacks. The comprehensive block diagram of the algorithm design is presented in Fig 2.

Take the matrix as an example to encrypt a single channel, and the specific encryption process is shown in Fig 3.

### Initial value perturbation

Taking the image $P$ of size $H \times W \times 3$ as an example, the eigenvalue of the image is read, and the MD5 hash function is used to generate the hexadecimal scrambling values $m(x)$, $x \in [1, 32]$, and processed by Eq (3) to generate the key parameters. This is the specific equation definition:

$$\begin{cases} a = 0.5 + (m(1) + m(10) + m(19) + m(28))/1000 \\ b = 2 + (m(2) + m(11) + m(20) + m(29))/1000 \\ x_1(0) = 0.1 + (m(3) + m(12) + m(21) + m(30))/1000 \\ x_2(0) = 0.1 + (m(4) + m(13) + m(22) + m(31))/1000 \end{cases} \qquad (3)$$

where $a$ and $b$ are key parameters, $x_1(0)$ and $x_2(0)$ are initial values of chaotic sequence.

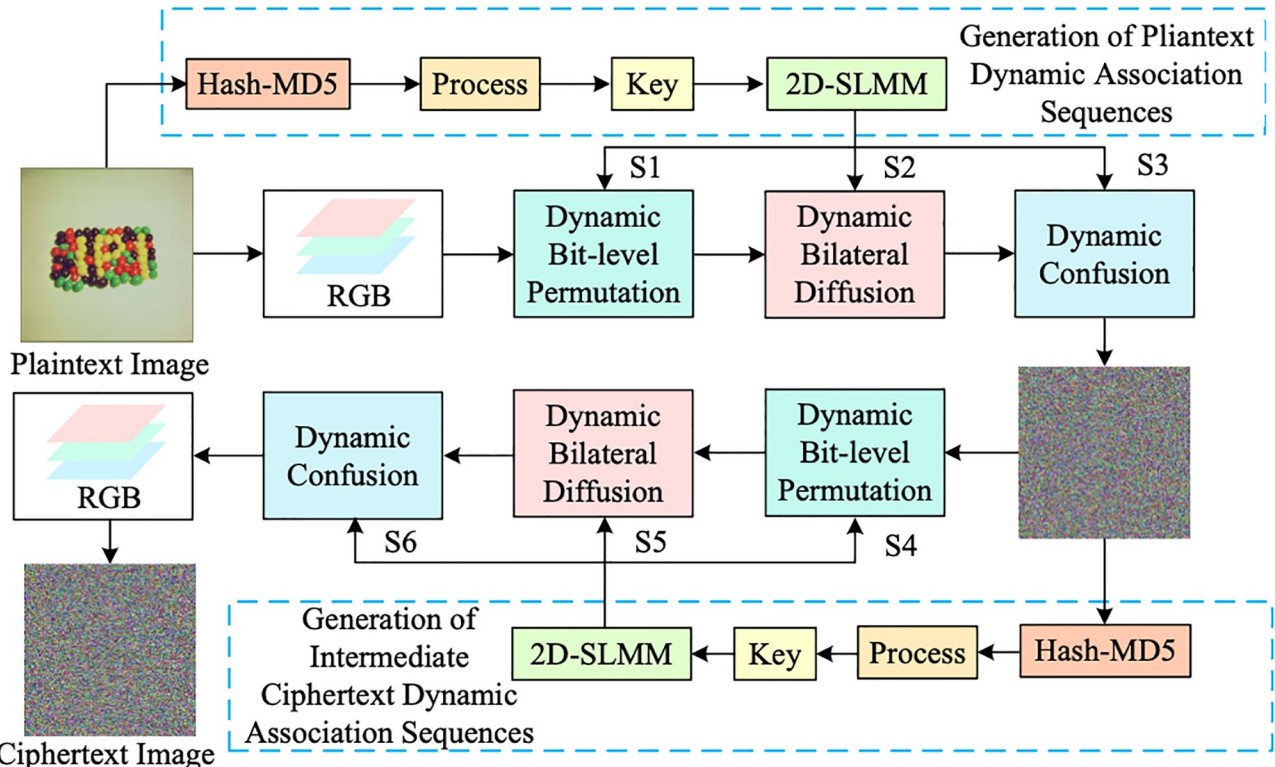

**Fig 2. Block diagram of the overall design of the cryptosystem.**

Whereby, the initial values of *a* and *b* are set to 0.5 and 2, respectively, to ensure their variation remains within the parameter range.

### Encryption process

Step 1: Sequence Generation and Preprocessing

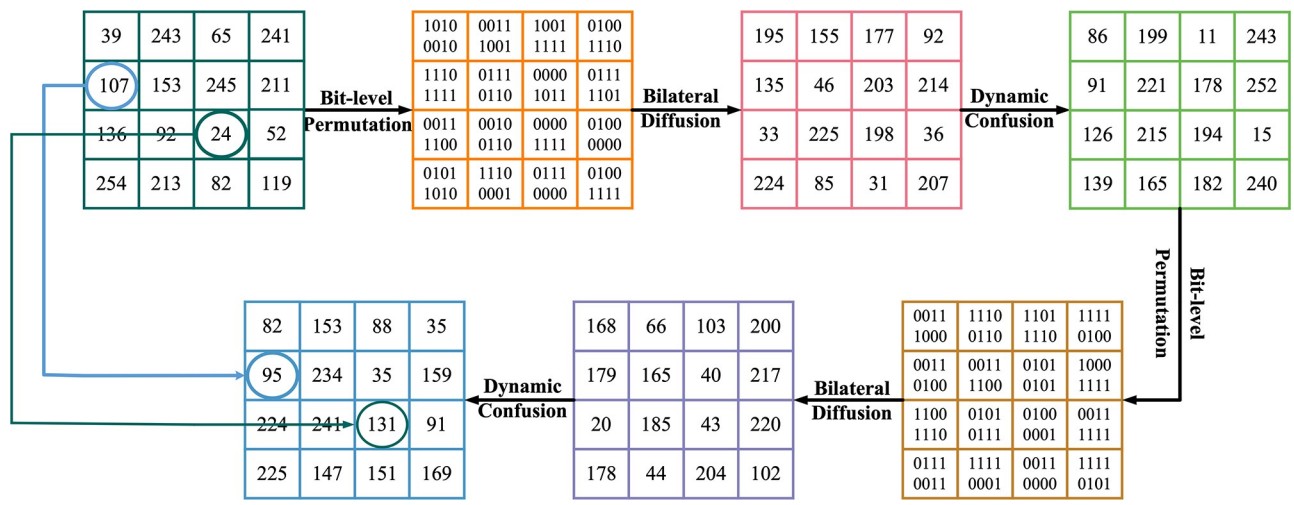

**Fig 3. Single-channel encryption example.**

The keys $a$, $b$, $x_1(0)$ and $x_2(0)$ generated based on the plaintext image are substituted into the 2D-SLMM chaotic system to generate three chaotic sequences $S_1$, $S_2$ and $S_3$, and the sequences $S_2$ and $S_3$ are processed by Eq (4), so that they are suitable for the sequences required by the algorithm. The explicit equation definition is as follows:

$$\begin{cases} S_2 = floor(S_2 \times 10^{15}) \bmod 256 \\ S_3 = floor(S_3 \times 10^{15}) \bmod 256 \end{cases} \tag{4}$$

where $floor(\cdot)$ is the downward integer function and $\bmod(\cdot)$ is the modulo operation function.

Step 2: Bit-level Permutation

The chaotic sequence $S_1$ is used to scramble the bit row and column of the plaintext image. Firstly, the sequence $S_1$ is sorted, and two sorting sequences $RK$ and $CK$ are obtained. Then, the image is expanded at the bit level, and the image size is adjusted to $H \times 8W$. Finally, the sorted sequence is used to operate the layered image. The specific operation is processed by Eq (5). Here is the precise equation definition:

$$\begin{cases} [w_1, RK] = sort(S_1(1:H)) \\ w_2, CK] = sort(S_1(H+1, 8W)) \\ C_1(i, j) = P(RK(i), CK(j)) \end{cases} \tag{5}$$

where $sort(\cdot)$ function indicates that each bit in the input sequence is sorted from low to high. $w_1$, $w_2$ represents the result of reordering the sequence. $i, j$ denote the sort index, $i = 1, 2, 3, \ldots, H, j = 1, 2, 3, \ldots, W$. $C_1$ is the image after bit-level permutation.

Step 3: Bilateral Diffusion

Using the pre-processed pseudo-random sequence $S_2$, and setting the first pixel $C_2(0) = 0$ of the ciphertext image, then calculating the sum of $C_2(0)$, $S_2(0)$, and $C_1(1)$, and performing modular operations on the sum to obtain the second cipher pixel $C_2(1)$. Where $C_1(1)$ is the first pixel to be encrypted in the permuted image, and $S_2(0)$ is the first pixel of the pseudo-random sequence $S_2$. The encrypted pixel $C_2(1)$ can be obtained by iterative relationship. In order to make the plaintext information completely hidden in the password pixels, reverse diffusion is needed to obtain the image $C_2'$ after bilateral diffusion. The specific formulas of the two diffusions can be represented by Eq (6). The equation's specific definition is outlined below:

$$\begin{cases} C_2(i) = (C_2(i-1) + S_2(i-1) + C_1(i)) \bmod 256 & i \in [1, H \times W] \\ C_2'(i) = (C_2(i-1) + S_2(i-1) + C_1(i)) \bmod 256 & i \in [H \times W, 1] \end{cases} \tag{6}$$

where $\bmod(\cdot)$ is the modulo operation function, $C_2$ is the image after the forward diffusion of the modulus, and $C_2'$ is the image after the reverse diffusion.

Step 4: Dynamic Confusion

Dynamic confusion allows each pixel to interact to achieve the avalanche effect. The pseudo-random sequence $S_3$ is used to generate two key matrices $K_{d1}$ and $K_{d2}$, and then the diffusion path is established between the pixels and the key matrices $K_{d1}$ and $K_{d2}$ are added to the diffusion process. The ciphertext pixels diffuse along this path to other pixels to generate an intermediate ciphertext image $C_3$. The confusion encryption generation equation of the first pixel value $C_3(1)$ of the image $C_3$ after dynamic confusion is defined by Eq (7). The specific

definition of the equation is given below:

$$\begin{cases} sum(1) = \sum_{i=1}^{L} C_2{}'(i) \\ \\ C_3(1) = C_2{}'(1) \oplus K_{d1}(1) \oplus (sum(1) \dot{+} K_{d2}(1)) \end{cases} \qquad (7)$$

where the operation symbol $\dot{}$ is defined as $a\dot{b} \triangleq (a+b) \bmod 256$. $K_{d1}$ and $K_{d2}$ are the key matrices composed of diffusion encryption sequences. The variable $sum$ represents the cumulative sum of all the pixels in the image $C_2$, and this sum is then used in an iterative process to generate the ciphertext image $C_3$. The specific formula is processed by Eq (8). The equation is defined in the following manner:

$$\{ C_3(i) = C_2{}'(i) \oplus (C_3(i-1) \dot{+} K_{d1}(i) \oplus (sum(i) \dot{+} K_{d2}(i))) sum(i) = sum(i-1) - C_2{}'(i) \ (8)$$

Step 5: Two-round Encryption of Intermediate Ciphertext

In order to enhance the ability of the algorithm to resist the chosen ciphertext attack, this paper innovatively refers to the ciphertext feedback mechanism, and uses the intermediate ciphertext image $C_3$ to generate the key through the MD5 hash function. Then the key is substituted into the 2D-SLMM chaotic system to generate a chaotic sequence. Finally, the sequence is used to perform bit-level permutation, bilateral diffusion and dynamic confusion on the intermediate ciphertext image $C_3$ to obtain the final ciphertext image $C_4$.

## Embedding a mask image

To convert a random ciphertext image into a meaningful output image, a DWT is used in the proposed of the proposed encryption scheme. The masking of the new image onto the the ciphertext image according to the following steps:

1) Take a mask image having meaningful information of size $2M \times 2N \times 3$.

2) Apply DWT to each color component of a mask image and extract four frequency sub-bands.

3) Now, split each pixel value of the pre-ciphertext image into its groups: (a) $LSB$-group and (b) $MSB$-group. For example, a pixel value having a grayscale value equal to 152 ($Gray_{dec}$ = 152), its binary version will be $Gray_{bin}$ = 10011000. The $LSB$ and $MSB$ group of the binary value will be $G_1$ = 1001 and $G_2$ = 1000, respectively.

4) Operate on each pixel until it reaches the position ($M, N$) for each color component. The $LSB$-group ($L - G$) and $MSB$-group ($M - G$) matrices are given in Eqs (9) and (10):

$$L - G = \begin{bmatrix} (01010000)_{1,1} & \cdots & (11100000)_{1,N} \\ (11000000)_{2,1} & \cdots & (10100000)_{2,N} \\ \vdots & \ddots & \vdots \\ (11000000)_{M-1,1} & \cdots & (10110000)_{M-1,N-1} \\ (10110000)_{M,1} & \cdots & (11100000)_{M,N} \end{bmatrix} \qquad (9)$$

$$M_G = \begin{bmatrix} (00001110)_{1,1} & \cdots & (00001000)_{1,N} \\ (00001100)_{2,1} & \cdots & (00001000)_{2,N} \\ \vdots & \ddots & \vdots \\ (00001110)_{M-1,1} & \cdots & (00001111)_{M-1,N-1} \\ (00001111)_{M,1} & \cdots & (00001000)_{M,N} \end{bmatrix} \tag{10}$$

5) The extracted high-frequency sub-bands (HL and HH) will be replaced with the two binary groups ($L - G$ and $M - G$).

6) After replacing the sub-bands, take the inverse DWT (IDWT) to restore the original mask image ($I_{R_{mask}}$). This $I_{R_{mask}}$ image will be transmitted as a meaningful encrypted image. The block diagram of the proposed embedding process is displayed in Fig 4.

## Decryption process

The decryption process is the inverse operation of encryption. Firstly, the sequence required for decryption is generated by the 2D-LSMM chaotic system. Then, the final ciphertext image is subjected to inverse dynamic confusion, inverse bilateral diffusion and inverse bit-level permutation to obtain the intermediate ciphertext image. Finally, after a round of decryption operation, the original image can be obtained. The specific decryption flow chart example is shown in the Fig 5.

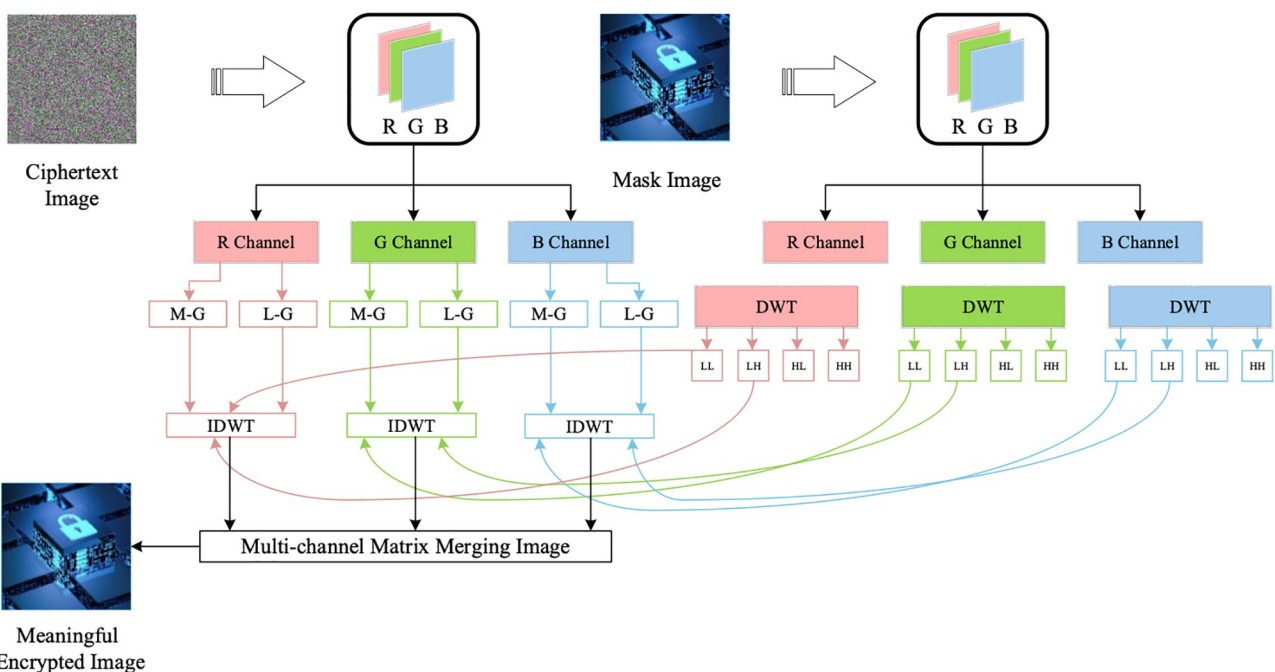

**Fig 4. Ciphertext image embedding process.**

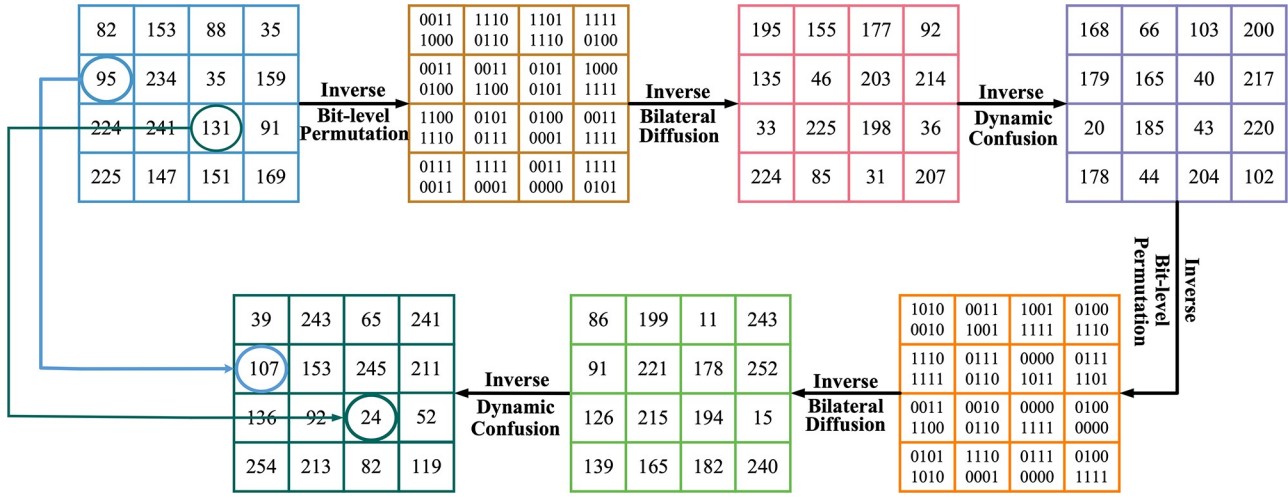

**Fig 5. Single-channel decryption example.**

## Experimental results and analysis discussion

### Experimental environment

We utilized a personal computer as the experimental platform, equipped with MATLAB R2023a experimental software. The PC is powered by an AMD Ryzen™ 7 CPU running at 3.2 GHz, with 32GB of memory and a 512GB hard disk. The operating system employed is Windows 10. Additionally, we selected USC-SIPI as the image data for the experiment.

### Cryptographic attacks

In the realm of cryptography, the well-established Kerckhoffs' principle asserts that the encryption algorithm of a secure cryptosystem should be openly known to attackers, with only the key remaining unknown [50]. The four commonly employed cryptanalysis methods, ranging from weak to strong, are as follows:

- **Ciphertext-only attack**
  Assuming that the attacker has access to only some ciphertext, their objective is to deduce the plaintext or the encryption key by analyzing the statistical properties of the ciphertext.

- **Known-plaintext attack**
  Assuming the attacker possesses partial plaintext and the corresponding ciphertext, their objective is to decipher or crack the associated encryption key and encryption algorithm.

- **Chosen-plaintext attack**
  Assuming the attacker has temporary access to the encryption machine, they can select plaintext that facilitates deciphering, obtain the corresponding ciphertext, and then launch an attack against the target algorithm.

- **Chosen-ciphertext attack**
  Assuming the attacker has temporary access to the decryption machine, they can select ciphertext that facilitates deciphering, obtain the corresponding plaintext, and then launch an attack against the target algorithm.

In the analysis of color image cryptosystems, chosen plaintext attacks and chosen ciphertext attacks are considered the most potent methods of cryptographic attack. The primary concept behind these attacks involves the careful selection of specific attack images, such as all-black or all-white images, followed by the application of algebraic analysis to deduce the original cryptosystem's equivalent key. In this paper, for instance, we utilize the digital image sets depicted in Fig 6(a)–6(d) and 6(m)–6(p) to decipher the target algorithm. Utilizing a comparable attack strategy, we undertake a security assessment of our proposed image encryption algorithm.

Taking the chosen-plaintext attack as an example, we designate Fig 6(a)–6(d) as the plaintext attack images, with the corresponding intermediate and final password images presented in Fig 6(e)–6(h). As demonstrated in Fig 6(i)–6(l), their histogram characteristics exhibit a noise-like pattern, which differs significantly from Fig 6(a)–6(d). To ensure generality, we also select the specific plaintext Fig 6(m)–6(p), with its corresponding ciphertext images displayed in Fig 6(q)–6(t). Also, the histogram plots in Fig 6(u)–6(x), for these ciphertexts exhibit a noisy pattern and differ significantly from the original text. Consequently, this approach makes it challenging for attackers to carry out penetration attacks. Similarly, achieving selective ciphertext attacks is also difficult.

The main source of these challenges arises from the incorporation of the diffusion-permutation-diffusion structure and the plaintext correlation mechanism. Concurrently, the use of the diffusion-permutation-diffusion structure and the intermediate ciphertext correlation mechanism effectively bolsters the algorithm's avalanche effect and security.

## Robustness analysis

This section aims to experimentally examine the robustness of the decryption algorithm against noise interference and image cropping attacks, while providing a detailed analysis of the algorithm's performance. This experiment involves introducing salt-and-pepper noise interference and image cropping attacks to encrypted ciphertext images, with the intention of simulating potential sources of interference in the decryption process. After subjecting the ciphertext to these attacks, the decryption process is initiated to assess whether the algorithm can effectively recover the compromised ciphertext images. Furthermore, the performance and stability of the algorithm in the face of noise interference and image cropping attacks will be evaluated.

**Salt-and-pepper noise analysis.**   In this experiment, an analysis of salt-and-pepper noise was conducted on the encrypted ciphertext images, with the objective of assessing the influence of varying intensities of salt-and-pepper noise on the decryption algorithm. This study explored three distinct levels of salt-and-pepper noise, namely 0.005, 0.01, and 0.03, to evaluate the performance of ciphertext recovery. The test results are depicted in Fig 7. The experimental findings indicate that the encryption algorithm exhibits excellent noise resistance.

**Cropping attack analysis.**   In this experiment, different degrees of cropping were applied to the ciphertext images, followed by decryption and subsequent analysis of the cropped ciphertext images. The experimental results are presented in Fig 8. The research findings demonstrate that despite the information cropping undergone by the ciphertext images prior to decryption, the content of the images remains clear and recognizable. This discovery highlights the excellent robustness of the proposed decryption algorithm when faced with information cropping attacks, successfully restoring the ciphertext images.

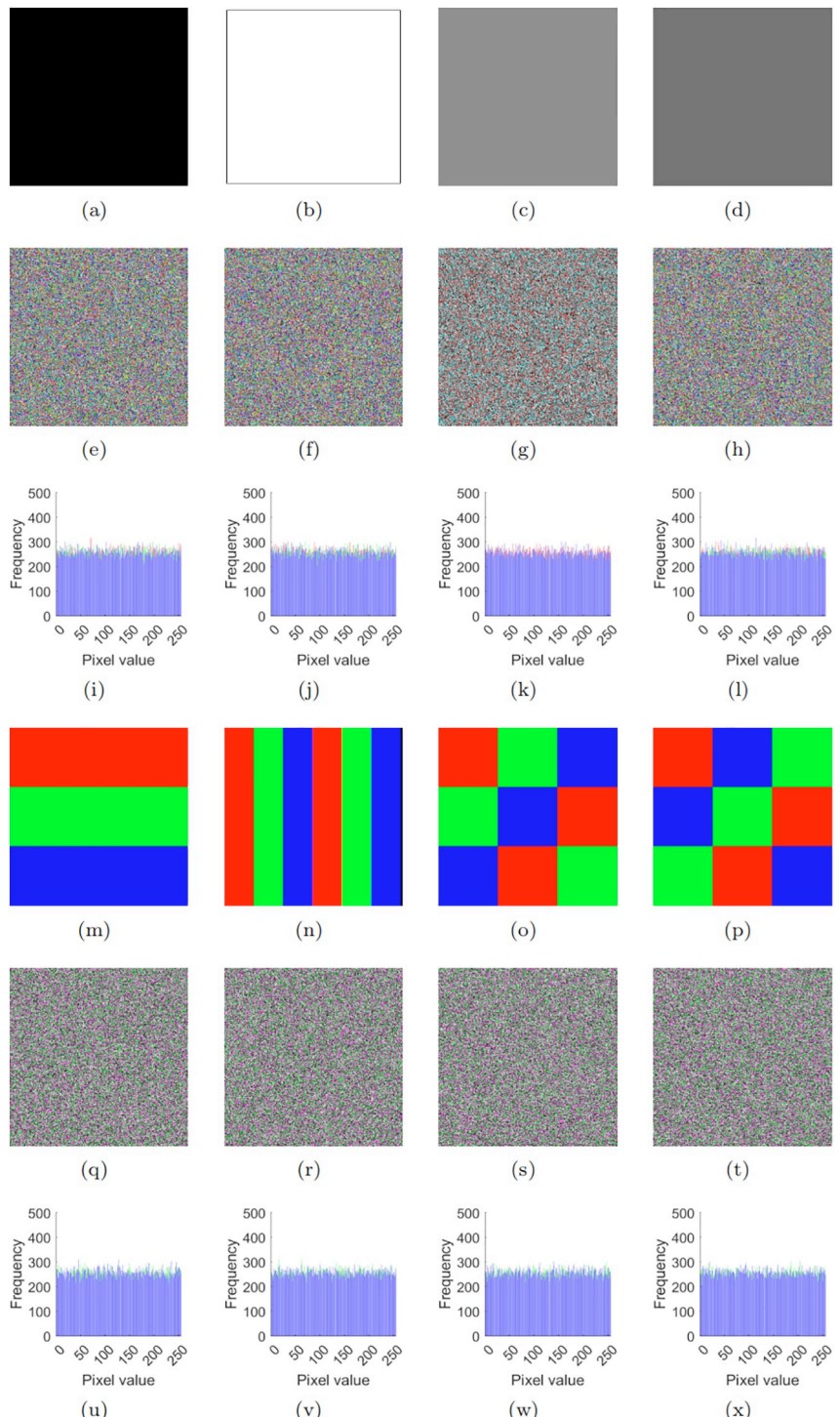

**Fig 6. Specific chosen plaintext images and their corresponding attack outcomes.**

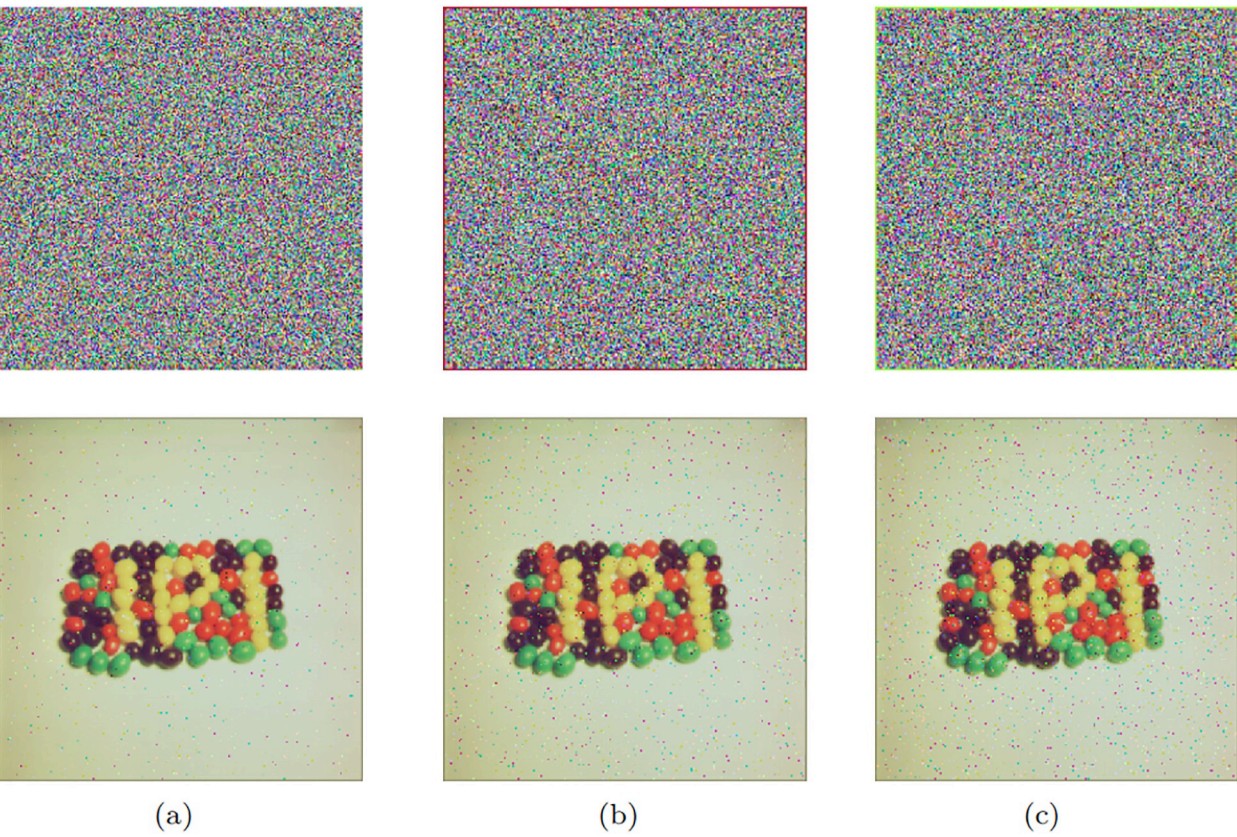

**Fig 7.** Salt and pepper noise test: (a) 0.005 noise; (b) 0.01 noise; (c) 0.015 noise.

## Experimental results and analysis

**Statistics histogram.** The histogram serves as a visual representation of the grayscale levels and their respective frequencies in an image. Typically, the histogram of a plaintext image showcases specific statistical patterns, whereas the histogram of an encrypted image portrays a distribution resembling random noise. Hence, a superior encryption algorithm can effectively transform the image into a distribution resembling noise, effectively concealing the essential information within the image. In Fig 9, we present the 3-D visual histograms of both plaintext and ciphertext channels. It is evident that the encrypted image effectively conceals the vital information present in the plaintext image, diminishing the likelihood of attackers decrypting the ciphertext image through statistical analysis. This ensures that adversaries are practically unable to reconstruct the original image based on the statistical characteristics of the encrypted image, thus preventing them from obtaining any valuable information.

**The coefficient of adjacent pixels.** A high degree of correlation among neighboring pixels suggests that the plaintext image is vulnerable to statistical analysis. Therefore, it is essential to reduce the correlation between adjacent pixels. Exceptional encryption algorithms can effectively lower the correlation among neighboring pixels. For our analysis, we randomly selected 3000 pixels from both the plaintext and ciphertext images and computed the horizontal, vertical, diagonal, and anti-diagonal correlations between adjacent pixels using Eq (11). The

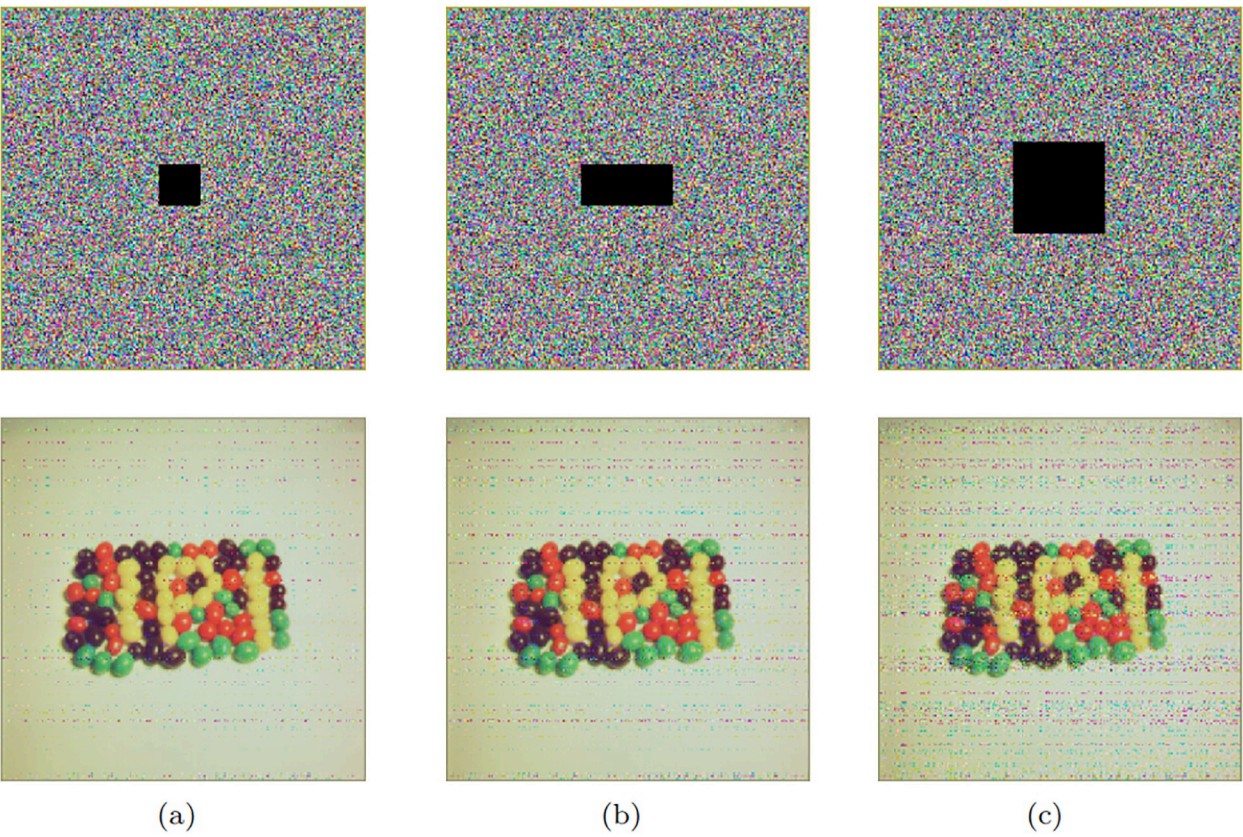

**Fig 8.** Cropping attack test: (a) Cropping of 32 × 32; (b) Cropping of 32 × 64; (c) Cropping of 64 × 64.

formula is defined as follows:

$$
\begin{cases}
r_{xy} = \dfrac{cov(x,y)}{\sqrt{D(x)}\sqrt{D(y)}} \\[2mm]
cov(x,y) = \frac{1}{N}\displaystyle\sum_{i=1}^{N}(x_i - E(x))(y_i - E(y)) \\[2mm]
D(x) = \frac{1}{N}\displaystyle\sum_{i=1}^{N}(x_i - E(x))^2 \\[2mm]
E(x) = \frac{1}{N}\displaystyle\sum_{i=1}^{N}x_i
\end{cases}
\tag{11}
$$

where $x_i$ and $y_i$ constitute the i-th pair of horizontal/vertical/diagonal/anti-diagonal neighboring pixels, $N$ is the total number of horizontal/vertical/diagonal/anti-diagonal neighboring pixels, $cov(x, y)$ is the covariance between pixel values $x$ and $y$, $D(x)$ and $D(y)$ are the pixel value $x$ and pixel value $y$ mean-square error, $E(x)$ and $E(y)$ are the expected values of pixel value $x$ and pixel value $y$, respectively. $r_{xy}$ is the correlation coefficient of pixel values $x$ and $y$. Fig 10 illustrates the correlations between pixels in different directions for the "Lena" image before and after encryption. It is apparent from the experimental data presented in Table 2 that the correlation coefficients for regular images are close to 1, whereas the correlation coefficients for

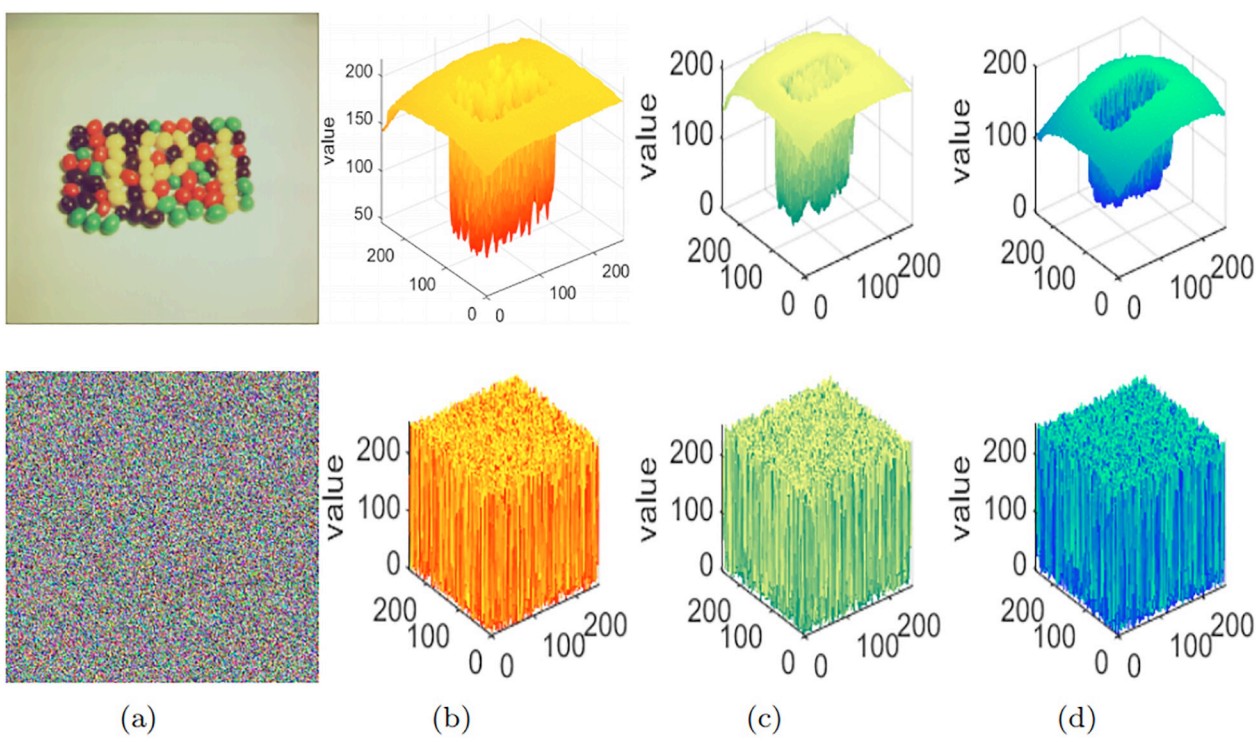

**Fig 9.** 3-D histogram comparison: (a) Plaintext image and ciphertext image; (b) R channel; (c) G channel; (d) B channel.

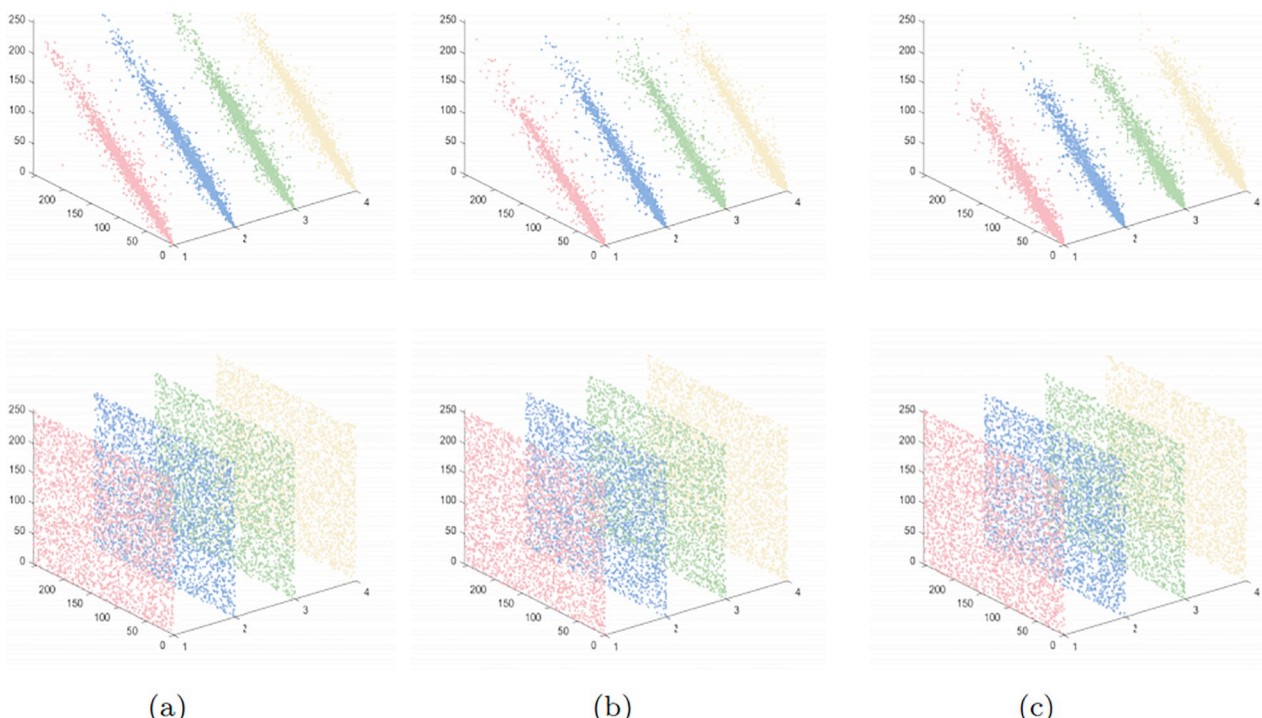

**Fig 10.** Comparing pixel correlations in different directions before and after encryption: (a) R channel; (b) G channel; (c) B channel.

**Table 2. The comparison results of the correlation coefficients of adjacent pixels.**

| Component | Direction | Original Image | The proposed | Ref. [51] | Ref. [52] |
|---|---|---|---|---|---|
| R channel | Horizontal | 0.9607 | -0.0039 | 0.0090 | -0.0014 |
| | Vertical | 0.9746 | 0.0129 | -0.0013 | 0.0011 |
| | Diagonal | 0.9476 | 0.0143 | -0.0025 | -0.0019 |
| | Anti-diagonal | 0.9434 | -0.0088 | - | - |
| G channel | Horizontal | 0.9653 | 0.0136 | -0.0027 | -0.0012 |
| | Vertical | 0.9705 | 0.0141 | -0.0051 | -0.0097 |
| | Diagonal | 0.9510 | 0.0106 | -0.0103 | -0.0045 |
| | Anti-diagonal | 0.9334 | 0.0006 | - | - |
| B channel | Horizontal | 0.9487 | 0.0103 | -0.0155 | -0.0058 |
| | Vertical | 0.9598 | 0.0271 | -0.0078 | 0.0063 |
| | Diagonal | 0.9331 | -0.0181 | 0.0099 | -0.0047 |
| | Anti-diagonal | 0.9330 | -0.0426 | - | - |

encrypted images are close to 0. This underscores that the approach proposed in this paper exhibits robust resistance to statistical attacks.

**Differential statistical analysis.** Differential attack is a common type of plaintext attack in which the attacker initially introduces minor alterations to a plaintext image and subsequently encrypts both variations of the plaintext image individually. Ultimately, the attacker gains information by comparing these two encrypted images. Normalized Pixel Change Rate (NPCR) and Unified Average Changing Intensity (UACI) are key metrics for evaluating the effectiveness of a differential attack. Their calculation formulas are defined as Eq (12), as follows:

$$
\begin{cases}
NPCR = \dfrac{1}{H \times W} \times \displaystyle\sum_{i=1}^{H}\sum_{j=1}^{W} D(i,j) \times 100\% \\[2ex]
UACI = \dfrac{1}{H \times W} \times \displaystyle\sum_{i=1}^{H}\sum_{j=1}^{W} \dfrac{|v_1(i,j) - v_2(i,j)|}{255} \times 100\%
\end{cases}
\tag{12}
$$

where $H \times W$ is the size of the image, and $v_1, v_2$ are the ciphertext images before and after changing one pixel of the plaintext image, respectively. $D$ can be defined by Eq (13):

$$
\begin{cases}
0 & if \quad v_1(i,j) = v_2(i,j) \\
1 & if \quad v_1(i,j) \neq v_2(i,j)
\end{cases}
\tag{13}
$$

Tables 3 and 4 shows the results of the algorithm calculated according to Eq (12). Fig 11 illustrates that even a minor disparity in the plaintext image when using the encryption scheme

**Table 3. NPCR test values.**

| Images | R channel | G channel | B channel | Mean |
|---|---|---|---|---|
| 4.1.01 | 99.5789 | 99.6216 | 99.6063 | 99.6022 |
| 4.1.04 | 99.6170 | 99.5834 | 99.6014 | 99.6006 |
| 4.1.05 | 99.6246 | 99.6262 | 99.6106 | 99.6205 |
| 4.1.06 | 99.5758 | 99.6109 | 99.6536 | 99.6134 |
| 4.2.07 | 99.6140 | 99.6143 | 99.5941 | 99.6075 |

**Table 4. UACI test values.**

| Images | R channel | G channel | B channel | Mean |
|---|---|---|---|---|
| 4.1.01 | 33.4578 | 33.4384 | 33.5079 | 33.4680 |
| 4.1.04 | 33.3441 | 33.4426 | 33.4776 | 33.4214 |
| 4.1.05 | 33.5127 | 33.4618 | 33.4126 | 33.4624 |
| 4.1.06 | 33.5009 | 33.3863 | 33.4604 | 33.4492 |
| 4.2.07 | 33.4255 | 33.4674 | 33.4972 | 33.5634 |

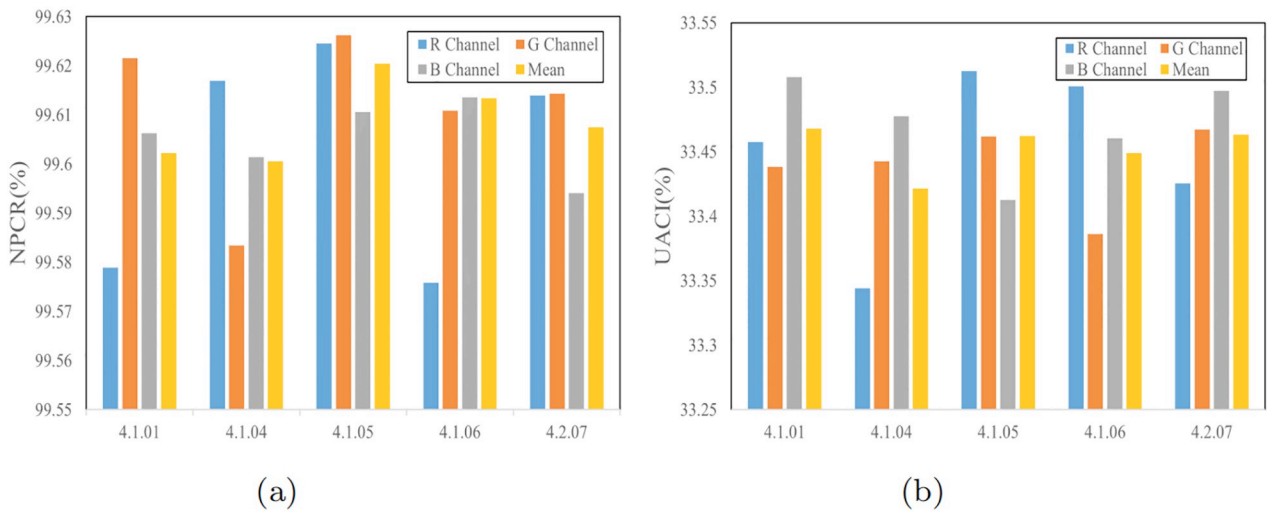

**Fig 11.** Test values of plaintext and ciphertext: (a) NPCR test values; (b) UACI test values.

proposed in this paper can yield outcomes that deviate significantly from the ideal. Consequently, it can be inferred that the encryption scheme presented in this paper exhibits strong resistance to differential attacks.

## Information entropy

The entropy value is used to assess the randomness and disorderliness of an information source, and when the entropy value approaches 8, it indicates a more random distribution of pixel values. We conducted a comparison of the information entropy of images before and after encryption, and the experimental results are presented in Table 5. It is evident from Table 5 that the experimental results are approximately close to 8, indicating that the proposed algorithm possesses favorable information entropy characteristics. The calculation method for the information entropy $H(m)$ of the information source $m$ is as follows:

$$H(m) = -\sum_{i=1}^{L} p(m_i) \log_2 p(m_i) \tag{14}$$

**Table 5. Information entropy test.**

| Images | 4.1.01 | 4.1.04 | 4.1.05 | 4.1.06 | 4.2.07 |
|---|---|---|---|---|---|
| Plain image | 6.8981 | 7.4270 | 7.0686 | 7.5371 | 7.6698 |
| Cipher image | 7.9998 | 7.9993 | 7.9991 | 7.9994 | 7.9998 |

where $L$ is the total number of symbols $m(i) \in m$ and $p(m_i)$ denotes the probability of the symbols.

## Key space analysis

The size of the key space plays a crucial role in determining the encryption algorithm's resilience against brute-force attacks. It is determined by the length of the secure key and is a critical feature in determining the strength of a cryptographic system. The image encryption algorithm designed in this paper uses a 2D-SLMM chaotic system, whose key space can be represented as $S \in \{x_1, x_2, a, b, MD5\}$, where $x_1, x_2, a, b$ are the key parameter with the precision of $10^{-16}$ and $MD5$ is the hash value introduced to enhance the key space, which can generate a 128bit hash value. Upon calculation, the key space size of this encryption scheme is approximately $10^{4 \times 16} \times 2^{128} \approx 2^{340}$, surpassing the $2^{100}$ requirement of the cryptographic system. This highlights that the algorithm presented in this paper can effectively withstand aggressive attacks. The key space comparison is detailed in Table 6.

## Sensitivity analysis

In this section, we evaluate the algorithm's performance using two criteria: key sensitivity and plaintext sensitivity. A robust algorithm should demonstrate a high degree of sensitivity, meaning that even slight modifications in either the encryption key or the plaintext image data during the encryption or decryption process can significantly influence the resulting encryption results.

**Analysis of sensitivity to the key.** Key sensitivity is assessed by encrypting the same image using two slightly different keys and analyzing the resulting ciphertexts. In this section, we encrypt the plaintext image using the original key, referred to as *key*, and a perturbed key, denoted as $key + 10^{-14}$. Subsequently, we assess the disparity between the resulting encrypted ciphertexts by computing the NPCR and UACI, as defined in Eq (12). The results are presented in Table 7 and Fig 12. It is evident that introducing a slight perturbation to the key leads to a significant difference between the two ciphertext images, with their NPCR and UACI values closely approaching the ideal values of 99.6094% and 33.4635%.

**Table 6. Key space comparison.**

| The proposed | Ref. [53] | Ref. [54] | Ref. [55] | Ref. [56] |
|---|---|---|---|---|
| $2^{340}$ | $2^{186}$ | $2^{168}$ | $2^{154}$ | $2^{224}$ |

**Table 7. Key sensitivity test of NPCR and UACI.**

| Images | Test | $a + 10^{-14}$ | $b + 10^{-14}$ | $h(0) + 10^{-14}$ | $w(0) + 10^{-14}$ |
|---|---|---|---|---|---|
| 4.1.01 | NPCR | 99.6078 | 99.6155 | 99.6033 | 99.6109 |
| 4.1.04 | | 99.6190 | 99.5972 | 99.6102 | 99.5941 |
| 4.1.05 | | 99.5936 | 99.6185 | 99.6160 | 99.6153 |
| 4.1.06 | | 99.5995 | 99.5895 | 99.6185 | 99.6124 |
| 4.2.07 | | 99.6243 | 99.6215 | 99.6096 | 99.6189 |
| 4.1.01 | UACI | 33.3953 | 33.4768 | 33.4038 | 33.4715 |
| 4.1.04 | | 33.4860 | 33.5115 | 33.4803 | 33.3996 |
| 4.1.05 | | 33.3810 | 33.3170 | 33.4269 | 33.4540 |
| 4.1.06 | | 33.4606 | 33.3518 | 33.4506 | 33.5103 |
| 4.2.07 | | 33.4379 | 33.4829 | 33.4327 | 33.4871 |

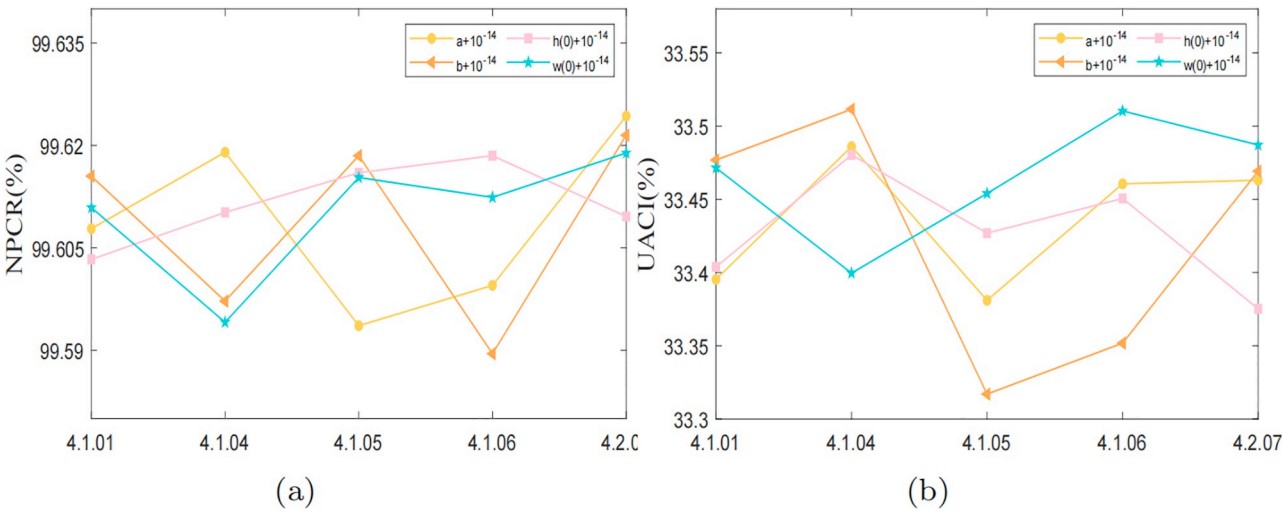

**Fig 12.** Key sensitivity test values: (a) NPCR test values; (b) UACI test values.

**Analysis of plaintext sensitivity.** Plaintext sensitivity gauges the degree of alteration in the associated ciphertext when the pixels of the plaintext are modified. When an algorithm lacks plaintext sensitivity, it becomes susceptible to decryption attempts that analyze discrepancies between plaintext and ciphertext pairs. Therefore, the algorithm's plaintext sensitivity is vital for its robustness against plaintext attacks. In this section, we assess the algorithm's sensitivity to alterations in the plaintext image by incrementing the pixel values at specific coordinates by 1: $(H/3, W/3)$, $(H/3, 2 \times W/3)$, $(2 \times H/3, W/3)$, and $(2 \times H/3, 2 \times W/3)$. We then compare the resulting NPCR and UACI values. The outcomes are presented in Table 8 and Fig 13. These results illustrate that the ciphertext image has undergone significant alterations, rendering it impractical for an attacker to undermine the algorithm through ciphertext comparison. Hence, the algorithm proposed in this paper demonstrates substantial resistance against plaintext attacks.

## Running efficiency

Operational efficiency stands as a pivotal marker for evaluating the feasibility and practicality of compression encryption algorithms. Algorithms with shorter runtimes are typically

**Table 8. The plaintext sensitivity test of NPCR and UACI.**

| Images | Test | (H/3, W/3) | (H/3, 2×W/3) | (2×H/3, W/3) | (2×H/3, 2 ×W/3) |
|---|---|---|---|---|---|
| 4.1.01 | NPCR | 99.6109 | 99.5992 | 99.6129 | 99.6262 |
| 4.1.04 | | 99.6165 | 99.6017 | 99.5926 | 99.6016 |
| 4.1.05 | | 99.6021 | 99.5972 | 99.6246 | 99.6160 |
| 4.1.06 | | 99.5941 | 99.6184 | 99.6231 | 99.6185 |
| 4.2.07 | | 99.6105 | 99.6082 | 99.6096 | 99.6040 |
| 4.1.01 | UACI | 33.4715 | 33.4209 | 33.4394 | 33.4476 |
| 4.1.04 | | 33.4161 | 33.4665 | 33.4715 | 33.4803 |
| 4.1.05 | | 33.4842 | 33.452 | 33.4787 | 33.4269 |
| 4.1.06 | | 33.4979 | 33.4979 | 33.4506 | 33.4506 |
| 4.2.07 | | 33.4379 | 33.4829 | 33.4327 | 33.4871 |

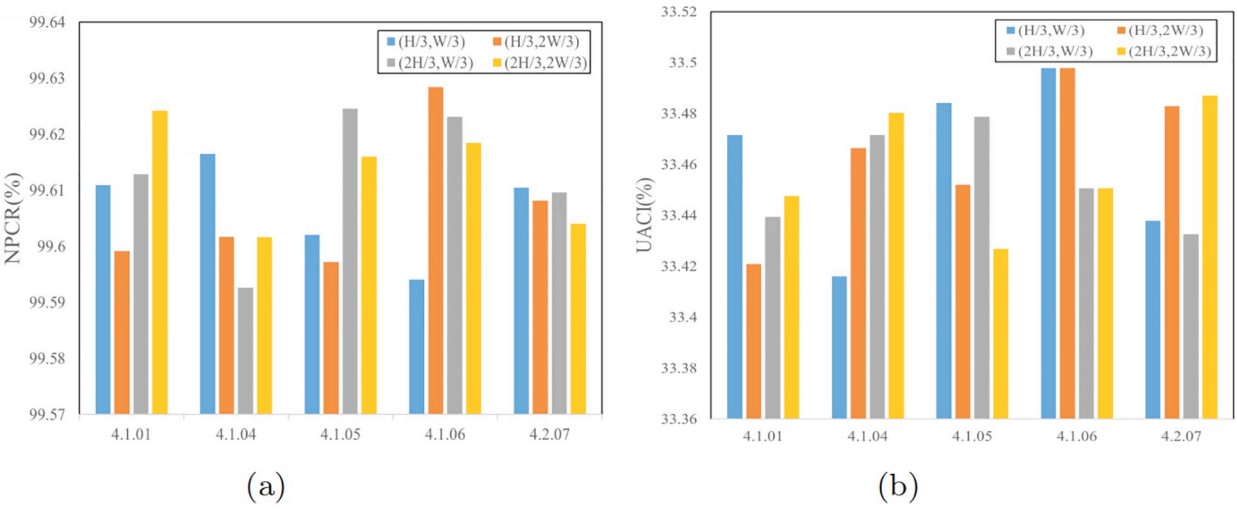

**Fig 13.** Plaintext sensitivity test values: (a) NPCR test values; (b) UACI test values.

**Table 9. Comparison results with some algorithms.**

| Algorithm | Permutation | Diffusion |
|---|---|---|
| **The proposed**(256 × 256) | **0.0359s** | **0.0267s** |
| **The proposed**(512 × 512) | **0.1205s** | **0.0919s** |
| **Ref**. [57](256 × 256) | 0.1084s | 0.0395s |
| **Ref**. [58](256 × 256) | 0.0380s | 0.0760s |
| **Ref**. [59](512 × 512) | 1.1918s | 0.2024s |
| **Ref**. [60](512 × 512) | 1.6800s | 0.1400s |

preferred. Within this experimental environment, we processed images sized at 256 × 256 and 512 × 512 individually, measuring the time consumption of two primary steps, bit-level permutation and bilateral diffusion. For comparative purposes, we referenced the operational efficiency from the literature, as demonstrated in Table 9. Compared to other algorithms, our findings indicate a leading speed in bit permutation and bidirectional diffusion, underscoring the effectiveness and satisfaction of our algorithm in practical terms. In addition, we have also conducted a comparative measurement of the algorithm's runtime against other studies. Table 10 shows that our algorithm possesses good encryption efficiency.

## Compression performance analysis

In the field of image processing, Peak Signal-to-Noise Ratio (PSNR) and Structural Similarity Index (SSIM) are two important metrics widely used for evaluating the quality of image encryption. PSNR involves the concept of Mean Squared Error (MSE), and its calculation is based on the inverse proportion of MSE to assess image quality. Specifically, MSE measures

**Table 10. Comparison on running time.**

| Image | The proposed | Ref. [57] | Ref. [54] | Ref. [58] |
|---|---|---|---|---|
| Lena (256 × 256) | **0.2743** | 0.4400 | 1.1168 | 1.2500 |

**Table 11. The test of PSNR and SSIM.**

| Images | PSNR | SSIM |
|--------|------|------|
| 4.1.01 | 37.9545 | 0.9676 |
| 4.1.04 | 36.9433 | 0.9627 |
| 4.1.05 | 39.6753 | 0.9708 |
| 4.1.06 | 37.5097 | 0.9654 |
| 4.2.07 | 35.8425 | 0.9587 |

**Table 12. Comparison of the average PSNR of different compression algorithms.**

| The proposed | Ref. [61] | Ref. [22] | Ref. [54] |
|--------------|-----------|-----------|-----------|
| 35.8425 | 32.0897 | 39.6445 | 34.1119 |

the average difference in pixel intensity between two images, while PSNR evaluates the ratio of this difference to the strength of the image signal, thereby providing a quantitative indicator of image quality loss. The MSE and PSNR are defined by:

$$\begin{cases} MSE = \dfrac{1}{H \times W} \sum_{i=1}^{H} \sum_{j=1}^{W} (P(i,j) - C(i,j))^2 \\ \\ PSNR = 10 \times \log_{10}\left(\dfrac{255^2}{MSE}\right) \end{cases} \tag{15}$$

where $H$ and $W$ are the height and width of the plaintext image $P$ and the ciphertext image $C$, while $Q$ represents the pixel level of the image. *SSIM* is a measure of the similarity between two images, defined as

$$SSIM(P, C) = \dfrac{(2\mu_p\mu_c + (0.01L)^2)(2\sigma_{pc} + (0.03L)^2)}{(\mu_p^2 + \mu_c^2 + (0.01L)^2)(\sigma_p^2 + \sigma_c^2 + (0.03L)^2)} \tag{16}$$

where $\mu_p$ and $\mu_c$ represent the mean value of the plaintext image $P$ and the ciphertext image $C$, respectively, $\sigma_p$ and $\sigma_c$ denote the standard deviation of the images $P$ and $C$, respectively, and $L$ represents the dynamic range of the pixel value. The values of PSNR and SSIM are calculated using the formulas mentioned above, and the specific experimental data are shown in Table 11. Compared with other algorithms, as shown in Table 12, the encryption effect of this scheme shows better performance.

## Conclusion

In this paper, a secure image communication scheme based on two-layer dynamic feedback encryption and discrete wavelet transform information hiding is proposed. The scheme introduces a dynamic feedback mechanism to ensure the security and confidentiality of the transmitted image, and implements it with the help of 2D-SLMM chaotic system combined with a variety of encryption technologies. By using various evaluation indicators to evaluate the algorithm, the experimental results fully prove the effectiveness of the scheme in resisting attacks and ensuring the confidentiality of transmitted images. DWT hidden cipher image also provides efficient encryption, higher levels of security and robustness to attacks. These encryption methods provide the scheme with excellent robust protection against various attacks and

ensure the confidentiality of transmitted images. From the experimental results, the secure image encryption scheme proposed in this paper provides good encryption results and will provide a reliable and innovative solution for securing transmitted images. Meanwhile, in the future, we will focus on the performance, security and scalability of the algorithm in practical applications, and seek further improvement and optimisation.

## Author Contributions

**Conceptualization:** Jinlong Zhang.

**Data curation:** Heping Wen.

**Formal analysis:** Heping Wen.

**Funding acquisition:** Heping Wen.

**Investigation:** Heping Wen.

**Methodology:** Jinlong Zhang.

**Project administration:** Heping Wen.

**Resources:** Heping Wen.

**Software:** Heping Wen.

**Supervision:** Heping Wen.

**Validation:** Heping Wen.

**Writing – original draft:** Jinlong Zhang, Heping Wen.

**Writing – review & editing:** Jinlong Zhang, Heping Wen.

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
