## [Decision Letter · Decision Letter 0]

15 Nov 2023

PONE-D-23-32782Secure image communication based on two-layer dynamic feedback encryption and DWT information hidingPLOS ONE

Dear Dr. Wen,

Thank you for submitting your manuscript to PLOS ONE. After careful consideration, we feel that it has merit but does not fully meet PLOS ONE’s publication criteria as it currently stands. Therefore, we invite you to submit a revised version of the manuscript that addresses the points raised during the review process.

We look forward to receiving your revised manuscript.

Kind regards,

Je Sen Teh

Academic Editor

PLOS ONE

4. We note that Figure(s) 2, 4, 6, 7, 8 and 9 in your submission contain copyrighted images. All PLOS content is published under the Creative Commons Attribution License (CC BY 4.0), which means that the manuscript, images, and Supporting Information files will be freely available online, and any third party is permitted to access, download, copy, distribute, and use these materials in any way, even commercially, with proper attribution. For more information, see our copyright guidelines: http://journals.plos.org/plosone/s/licenses-and-copyright.

a. You may seek permission from the original copyright holder of Figure(s) 2, 4, 6, 7, 8 and 9 to publish the content specifically under the CC BY 4.0 license. 

Additional Editor Comments:

The reviewers have highlighted some concerns about the manuscript that need to be addressed before it can be considered for publication.

Here are some general observations. Please check all reviewer comments in detail:

1. There are some inconsistencies and lack of clarity with some of the equations. Authors need to make corrections and provide clarifications.

2. The efficiency of the proposed encryption algorithm needs to be analysed.

3. Justifications for some design decisions (two-layer feedback vs one-layer) should be provided.

4. Comparison with the state-of-the-art must be included.

5. Rewriting abstract/conclusion to better highlight contributions, future work, etc.

Proofreading of the manuscript is also recommended.

Reviewers' comments:

Reviewer's Responses to Questions

**Comments to the Author**

1. Is the manuscript technically sound, and do the data support the conclusions?

Reviewer #1: Yes

Reviewer #2: Partly

2. Has the statistical analysis been performed appropriately and rigorously? 

Reviewer #1: Yes

Reviewer #2: Yes

3. Have the authors made all data underlying the findings in their manuscript fully available?

Reviewer #1: Yes

Reviewer #2: Yes

4. Is the manuscript presented in an intelligible fashion and written in standard English?

Reviewer #1: Yes

Reviewer #2: Yes

5. Review Comments to the Author

Reviewer #1: In this manuscript, the authors propose a secure image encryption scheme based on the 2D-SLMM chaotic system and a dynamic feedback mechanism. Overall, the manuscript is well-organized and well-written, and the work presented is both innovative and interesting. However, before accepting this manuscript for publication, the following minor issues need to be addressed:

1. Considering the length of the article, the authors are advised to carefully proofread the manuscript to remove any possible minor errors, such as typos.

2. The authors are advised to adjust the content of their abstract to more attractively present the motivations, methods, results, and conclusions of their work.

3. The manuscript comprises a few lengthy sentences, and the authors should contemplate modifying them to enhance readability.

4. It is recommended that Table 1 be appropriately adjusted to better present relevant content.

5. Similarly, it is recommended that the figures in the manuscript be reviewed and adjusted to better present relevant content.

6. The authors are advised to provide a more detailed explanation of the motivation, reasons, and purpose behind the utilization of the MD5 hash function.

7. In line 333, the variable i is not italicized. Therefore, it is strongly advised that the authors carefully review the entire text to avoid such oversights.

8. While considering security, encryption efficiency is also an important factor that researchers must consider. Therefore, it is recommended that the authors analyze and discuss the encryption efficiency of the proposed scheme.

9. It is recommended that the authors adjust and improve the content of the conclusion section.

10. Is there room for further refinement or improvement of the current work in the future? In other words, what is your future further work related to this manuscript?

11. The authors are advised to carefully proofread and standardize their references in accordance with the template.

Reviewer #2: The authors propose an image encryption method which uses DWT and two dynamic feedback encryptions (plaintext and intermediate ciphertext). The article describes the encryption and decryption process in detail and presents the experimental data well with images and tables.

Here are my comments:

1. Are a and b in equations (1) and (3) the same? If they are the same, why is the description of them inconsistent?

2. The author's description of equation (3) is vague. For example, why are initial values assigned to a and b: 0.5 and 2 respectively? m(x) is in decimal or hexadecimal? What is the significance of such data processing?

3. How efficient is the encryption and decryption?

4. What are the advantages of two-layer dynamic feedback encryption over single-layer? How effective is the algorithm if it uses only plaintext feedback or intermediate ciphertext feedback? Please add this aspect in the experimental section.

5. The experimental data needs to be compared with recent articles, and it is clear that some articles are not appropriate, e.g., Ref.[49] and Ref.[50].

6. Why is there no reference in the Sensitivity analysis section?

7. The original manuscript mentions four innovations, including the use of DWT, in the introduction, but not in the conclusion?

In addition, the manuscript needs to be revised in the following ways:

1. Show the full name of the 2D-SLMM chaotic system as early as possible, not in line 93.

2. Change the abbreviations of Figure and Equation. They should be Fig. and Eq..

3. In Figure 2, Seq1 should be S1, and the others should be the same.

4. Use abbreviation for Figure at line 273.

5. Unify the format of the literature.

6. The source of the pictures used in the experiment should be stated.

6. PLOS authors have the option to publish the peer review history of their article (what does this mean?). If published, this will include your full peer review and any attached files.

Reviewer #1: No

Reviewer #2: No

---

## [Author Response · Author response to Decision Letter 0]

29 Nov 2023

Response to Reviewer 1

[General Comment] In this manuscript, the authors propose a secure image encryption scheme based on the 2D-SLMM chaotic system and a dynamic feedback mechanism. Overall, the manuscript is well-organized and well-written, and the work presented is both innovative and interesting. However, before accepting this manuscript for publication, the following minor issues need to be addressed:

Response: Thank you very much for the valuable time you have allocated to review this manuscript. We are honored to receive your suggestions and are very willing to make improvements to this article!

[Comment 1] Considering the length of the article, the authors are advised to carefully proofread the manuscript to remove any possible minor errors, such as typos.

Response: Thank you for your suggestion. We have carefully reviewed the entire article and made necessary proofreading and modifications to eliminate any potential spelling and other minor errors.

[Comment 2] The authors are advised to adjust the content of their abstract to more attractively present the motivations, methods, results, and conclusions of their work.

Response: Thank you for the suggestion. We have adjusted the content of the abstract to more attractively present the motivations, methods, results, and conclusions of our work.

[Comment 3] The manuscript comprises a few lengthy sentences, and the authors should contemplate modifying them to enhance readability.

Response: Thank you for your valuable suggestion. We have carefully reviewed the issue you raised regarding lengthy sentences in the manuscript and have made corresponding modifications to enhance overall readability.

 [Comment 4] It is recommended that Table 1 be appropriately adjusted to better present relevant content. 

Response: Thank you for your suggestion. Table 1 has been appropriately adjusted to better present the relevant content. The modified table now provides information more clearly, facilitating better understanding for readers.

[Comment 5] Similarly, it is recommended that the figures in the manuscript be reviewed and adjusted to better present relevant content. 

Response: Thank you for your suggestion. We have thoroughly reviewed and adjusted the figures in the manuscript to better present the relevant content. Following these corrections, the figures now convey information more clearly, aiding in a better understanding of the paper's content. 

[Comment 6] The authors are advised to provide a more detailed explanation of the motivation, reasons, and purpose behind the utilization of the MD5 hash function. 

Response: The MD5 function, as a type of hashing algorithm, plays a crucial role in data security. It can transform data of any length into a fixed-length hash value and is widely utilized in data integrity verification, digital signatures, and encryption domains. In our study, the utilization of the MD5 function forms an intermediary ciphertext mechanism, enhancing the encryption performance of the data. This approach not only bolsters data security but also ensures the integrity and validation during data transmission. 

[Comment 7] In line 333, the variable i is not italicized. Therefore, it is strongly advised that the authors carefully review the entire text to avoid such oversights. 

Response: Thank you for your careful observation. We have conducted a thorough check of the entire manuscript and rectified similar issues, including the non-italicized variable "i" in line 333.

[Comment 8] While considering security, encryption efficiency is also an important factor that researchers must consider. Therefore, it is recommended that the authors analyze and discuss the encryption efficiency of the proposed scheme. 

Response: Thank you for the suggestion. We have conducted experiments and included an analysis of the encryption efficiency of the proposed scheme in our discussion. 

[Comment 9] It is recommended that the authors adjust and improve the content of the conclusion section. 

Response: Thank you for the suggestion. We have adjusted and improved the content of the conclusion section.

[Comment 10] Is there room for further refinement or improvement of the current work in the future? In other words, what is your future further work related to this manuscript? 

Response: In the future, we plan to apply the encryption algorithm proposed in this research to practical scenarios. Our focus will be on assessing the performance, security, and scalability of the algorithm in real-world applications, seeking further enhancements and optimizations.

[Comment 11] The authors are advised to carefully proofread and standardize their references in accordance with the template. 

Response: Thank you for the suggestion. We have made careful revisions and standardized the references according to the template requirements. 

Response to Reviewer 2

[General Comment] The authors propose an image encryption method which uses DWT and two dynamic feedback encryptions (plaintext and intermediate ciphertext). The article describes the encryption and decryption process in detail and presents the experimental data well with images and tables.

Response: Thank you very much for your previous comments that helped us improve this manuscript. We have carefully studied your suggestions and have tried our best to address each of these comments. We hope the manuscript has been improved accordingly.

[Comment 1] Are a and b in equations (1) and (3) the same? If they are the same, why is the description of them inconsistent? 

Response: Thank you for your inquiry. The 'a' and 'b' in equations (1) and (3) are not the same. We have updated the descriptions to ensure accuracy and consistency.

[Comment 2] The author's description of equation (3) is vague. For example, why are initial values assigned to a and b: 0.5 and 2 respectively? m(x) is in decimal or hexadecimal? What is the significance of such data processing? 

Response: Thank you for your attention. The initial values of 'a' and 'b' are restricted by the parameter range of the chaotic system. 'm(x)' is generated by the MD5 function and represented in hexadecimal. The significance of this data processing lies in its association with plaintext, enhancing resistance against chosen-plaintext attacks. We have revised the description in the manuscript to better articulate these concepts.

[Comment 3] How efficient is the encryption and decryption?

Response: Thank you for your inquiry. We have conducted experiments to assess the efficiency of encryption.

[Comment 4] What are the advantages of two-layer dynamic feedback encryption over single-layer? How effective is the algorithm if it uses only plaintext feedback or intermediate ciphertext feedback? Please add this aspect in the experimental section. 

Response: The two-layer dynamic feedback encryption offers advantages such as enhanced encryption performance and stronger resistance against chosen-ciphertext attacks compared to single-layer encryption. Manuscripts have been added for related experiments.

[Comment 5] The experimental data needs to be compared with recent articles, and it is clear that some articles are not appropriate, e.g., Ref.[49] and Ref.[50]. 

Response: Thank you for your guidance. We have revised the references to ensure the experimental data aligns with recent articles. 

[Comment 6] Why is there no reference in the Sensitivity analysis section?

Response: Thank you for your reminder. In the Sensitivity Analysis section, we primarily focused on exploring the inherent characteristics of the data and did not reference other literature. The data presented in this section shows results that are very close to the ideal values. 

[Comment 7] The original manuscript mentions four innovations, including the use of DWT, in the introduction, but not in the conclusion?

Response: Thank you for pointing that out. We have made modifications to the conclusion section to ensure it covers the innovations mentioned in the introduction, including the use of DWT. 

[Comment 8] In addition, the manuscript needs to be revised in the following ways:

1. Show the full name of the 2D-SLMM chaotic system as early as possible, not in line 93.

2. Change the abbreviations of Figure and Equation. They should be Fig. and Eq..

3. In Figure 2, Seq1 should be S1, and the others should be the same.

4. Use abbreviation for Figure at line 273.

5. Unify the format of the literature.

6. The source of the pictures used in the experiment should be stated.

Response: Thank you for the detailed guidance. We have thoroughly checked and made the following modifications to the manuscript:

1.We have presented the full name of the 2D-SLMM chaotic system at an earlier point, not limited to line 93.

2.The abbreviations for figures and equations have been changed to Fig. and Eq..

3.In Figure 2, 'Seq1' has been updated to 'S1', and the corresponding adjustments have been made for others.

4.The abbreviation for 'Figure' has been used at line 273.

5.We have standardized the format of the references.

6.The sources of the images used in the experiments have been indicated.

---

## [Decision Letter · Decision Letter 1]

17 Jan 2024

PONE-D-23-32782R1Secure image communication based on two-layer dynamic feedback encryption and DWT information hidingPLOS ONE

Dear Dr. Wen,

Thank you for submitting your manuscript to PLOS ONE. After careful consideration, we feel that it has merit but does not fully meet PLOS ONE’s publication criteria as it currently stands. Therefore, we invite you to submit a revised version of the manuscript that addresses the points raised during the review process.

One of the previous reviewers was satisfied with the changes made. However, as another reviewer was not responsive, we engaged a third reviewer to ensure that all modifications from the previous review were incorporated. Unfortunately, some additional concerns were highlighted that needs to be addressed. Please see my comments and the reviewer's comments below.

We look forward to receiving your revised manuscript.

Kind regards,

Je Sen Teh

Academic Editor

PLOS ONE

Additional Editor Comments:

The following concerns need to be addressed:

Providing justification for the use of hash functions to generate the secret key. An additional observation is that the cipher is not secure at all against a known-plaintext (or known-plainimage) attack because the secret key is dependent on the plainimage. Knowledge of the plainimage is thus equivalent to knowledge of the key. As the reviewer pointed out, this deviates from common principles of cryptography. Some discussion as to how such a scheme would work in a real world security protocol would be beneficial.Providing clarification about the permutation step.When measuring runtime, an average across multiple runs would be more accurate. Even better, an analysis of the number of primitive operations would be independent of the computing machine and provide a more fair comparison.Comparisons were made with cryptosystems that are not the current state of the art (2020-2021). More recent (2023/2024) cryptosystems should be used as a benchmark.

Reviewers' comments:

Reviewer's Responses to Questions

**Comments to the Author**

1. If the authors have adequately addressed your comments raised in a previous round of review and you feel that this manuscript is now acceptable for publication, you may indicate that here to bypass the “Comments to the Author” section, enter your conflict of interest statement in the “Confidential to Editor” section, and submit your "Accept" recommendation.

Reviewer #1: All comments have been addressed

Reviewer #3: (No Response)

2. Is the manuscript technically sound, and do the data support the conclusions?

Reviewer #1: Yes

Reviewer #3: Partly

3. Has the statistical analysis been performed appropriately and rigorously? 

Reviewer #1: Yes

Reviewer #3: Yes

4. Have the authors made all data underlying the findings in their manuscript fully available?

Reviewer #1: Yes

Reviewer #3: No

5. Is the manuscript presented in an intelligible fashion and written in standard English?

Reviewer #1: Yes

Reviewer #3: Yes

6. Review Comments to the Author

Reviewer #1: With the efforts of the authors, the quality of the paper has been significantly improved. Therefore, the paper can be accepted for publication.

Reviewer #3: The authors used MD5 to generate hash values and employed it to create the secret key, utilizing MD5 twice—once for round 1 and again for round 2. Firstly, the secret key should be larger than what they calculated because they will share 128 bits for each round with the receiver. I disagree with this approach as it deviates from the principles of cryptography.

The authors should calculate PSNR because they employed frequency transformation in the encryption algorithm. The permutation step is not clear; it's essential to specify whether it's a bit permutation. If it is, the speed of the permutation operation needs to be addressed, as it might take more time compared to pixel permutations. This aspect needs to be highlighted further.

2D chaotic maps require time, and I believe this is not included in the time calculation. Additionally, one measurement may not be sufficient for it. It would be beneficial to review some latest articles and expand the comparison to demonstrate the superiority of the paper.

****Manuscript suggestions by the reviewer have been redacted**. **Please identify recently proposed cryptosystems (2023/2024) to be used as benchmarks for the proposed work, especially for performance comparisons.***

7. PLOS authors have the option to publish the peer review history of their article (what does this mean?). If published, this will include your full peer review and any attached files.

Reviewer #1: No

Reviewer #3: **Yes: **Moatsum Alawida

---

## [Author Response · Author response to Decision Letter 1]

10 Feb 2024

All responses are in "Response. docx".

---

## [Decision Letter · Decision Letter 2]

26 Feb 2024

Secure image communication based on two-layer dynamic feedback encryption and DWT information hiding

PONE-D-23-32782R2

Dear Dr. Wen,

We’re pleased to inform you that your manuscript has been judged scientifically suitable for publication and will be formally accepted for publication once it meets all outstanding technical requirements.

Kind regards,

Je Sen Teh

Academic Editor

PLOS ONE

Additional Editor Comments (optional):

Reviewers' comments:

Reviewer's Responses to Questions

**Comments to the Author**

1. If the authors have adequately addressed your comments raised in a previous round of review and you feel that this manuscript is now acceptable for publication, you may indicate that here to bypass the “Comments to the Author” section, enter your conflict of interest statement in the “Confidential to Editor” section, and submit your "Accept" recommendation.

Reviewer #1: All comments have been addressed

Reviewer #3: All comments have been addressed

2. Is the manuscript technically sound, and do the data support the conclusions?

Reviewer #1: Yes

Reviewer #3: Yes

3. Has the statistical analysis been performed appropriately and rigorously? 

Reviewer #1: Yes

Reviewer #3: Yes

4. Have the authors made all data underlying the findings in their manuscript fully available?

Reviewer #1: Yes

Reviewer #3: Yes

5. Is the manuscript presented in an intelligible fashion and written in standard English?

Reviewer #1: Yes

Reviewer #3: Yes

6. Review Comments to the Author

Reviewer #1: I would like to express my sincere thanks to the authors for their revisions based on my review comments. With the efforts of the authors, the quality of the paper has been significantly improved. Therefore, the paper can be accepted for publication.

Reviewer #3: The authors assured me that they would address my first concern in the upcoming paper. Thank you for letting me know. As for this paper, I suggest adding justifications for the mentioned point. Overall, the paper requires only minor revisions.

7. PLOS authors have the option to publish the peer review history of their article (what does this mean?). If published, this will include your full peer review and any attached files.

Reviewer #1: No

Reviewer #3: No
